

# Overcoming the gender bias in ecology and evolution: is the double-anonymized peer review an effective pathway over time?

Cibele Cássia-Silva[1,2], Barbbara Silva Rocha[2,3], Luisa Fernanda Liévano-Latorre[2,4], Mariane Brom Sobreiro[2,5] and Luisa Maria Diele-Viegas[2,6]

[1] Department of Plant Biology/Institute of Biology, University of Campinas, Campinas, SP, Brazil
[2] Kunhã Asé Network of Women in Science, Salvador, Bahia, Brazil
[3] INRAE, Aix Marseille Université, UMR RECOVER, Aix-en-Provence, France
[4] International Institute for Sustainability, Rio de Janeiro, Rio de Janeiro, Brazil
[5] Central Public Health Laboratory of Goiás, Goiânia, Goiás, Brazil
[6] Laboratory of (Bio)Diversity in the Anthropocene/Institute of Biology, Federal University of Bahia, Salvador, Bahia, Brazil

Corresponding author
Cibele Cássia-Silva,
cibelecassia01@gmail.com

## ABSTRACT

Male researchers dominate scientific production in science, technology, engineering, and mathematics (STEM). However, potential mechanisms to avoid this gender imbalance remain poorly explored in STEM, including ecology and evolution areas. In the last decades, changes in the peer-review process towards double-anonymized (DA) have increased among ecology and evolution (EcoEvo) journals. Using comprehensive data on articles from 18 selected EcoEvo journals with an impact factor >1, we tested the effect of the DA peer-review process in female-leading (*i.e.*, first and senior authors) articles. We tested whether the representation of female-leading authors differs between double and single-anonymized (SA) peer-reviewed journals. Also, we tested if the adoption of the DA by previous SA journals has increased the representativeness of female-leading authors over time. We found that publications led by female authors did not differ between DA and SA journals. Moreover, female-leading articles did not increase after changes from SA to DA peer-review. Tackling female underrepresentation in science is a complex task requiring many interventions. Still, our results highlight that adopting the DA peer-review system alone could be insufficient in fostering gender equality in EcoEvo scientific publications. Ecologists and evolutionists understand how diversity is important to ecosystems' resilience in facing environmental changes. The question remaining is: why is it so difficult to promote and keep this "diversity" in addition to equity and inclusion in the academic environment? We thus argue that all scientists, mentors, and research centers must be engaged in promoting solutions to gender bias by fostering diversity, inclusion, and affirmative measures.

## INTRODUCTION

Females represent half of the global population but are still underrepresented in most work fields, mainly in politics and economics (*World Economic Forum, 2019*). In the education field, despite recent progress (*World Economic Forum, 2019*), we still need to reach equity. For instance, females are still strongly underrepresented in STEM (science, technology, engineering, and mathematics) graduate courses, and research positions, where only 33% of researchers are females (*Garcia-Holgado et al., 2020*; *UNESCO, 2021*).

Gender bias in the academic environment has multiple dimensions and causes manifested in society, home, workplace, and individuals (*Leite & Diele-Viegas, 2020*). Girls are discouraged from pursuing a STEM career from an early age due to gender stereotypes, which prevents many from even considering such careers as possible paths (*Leite & Diele-Viegas, 2020*). Following this initial cultural barrier, females in STEM careers may face additional obstacles in their professional development, such as biases regarding their abilities, harassment, and discrimination (*Pell, 1996*; *Greider et al., 2019*; *Huang et al., 2020*). A central motivation for such barriers is the tendency of male scientists to support, collaborate with, hire, and mentor male scientists (*Brashears, 2008*; *Moss-Racusin et al., 2012*; *Sheltzer & Smith, 2014*), a feature of human behavior known as gender homophily. The principle of homophily states that people with similar characteristics, such as gender, race, ethnicity, age, socioeconomic background, and educational attainment, tend to interact between them more frequently than those with different characteristics (*McPherson, Smith-Lovin & Cook, 2001*). Therefore, gender homophily can potentially lead to decreased participation and competitiveness of female researchers in the academic environment, particularly in leadership positions (*Sheltzer & Smith, 2014*). Indeed, female researchers represent only 12% of members of national science academies and 33.3% of STEM researchers, and they also receive fewer grants (*UNESCO, 2021*). Evidence of gender bias has also been found in authorship patterns in STEM scientific publications (*Fox, Ritchey & Paine, 2018*; *Holman, Stuart-Fox & Hauser, 2018*; *Salerno et al., 2019*).

A gender gap of 27% in total productivity and females representing less than 30% of authors indicate the pervasiveness of gender bias in scientific publications (*West et al., 2013*; *Larivière et al., 2013*; *Huang et al., 2020*). Specifically, in biological science publications, less than 40% of articles have females as first authors, and less than 30% have females as last (*i.e.*, senior) authors (*Bendels et al., 2018*; *Fox, Ritchey & Paine, 2018*; *Salerno et al., 2019*). In the field of ecology and evolution (EcoEvo), despite submitted articles with female authorships being sent to peer review at similar rates to articles with male authorships, females are significantly underrepresented as first, senior, and sole authors (*Fox & Paine, 2019*). Gender bias is also evident in corresponding authors, as articles with a female corresponding author have lower probabilities of a positive outcome, and female first authors tend to defer corresponding authorship to a coauthor (*Fox, Ritchey & Paine, 2018*; *Edwards, Schroeder & Dugdale, 2019*; *Fox & Paine, 2019*). Additionally, the proportion of female co-authors in EcoEvo is higher when the senior author is female (*Fox, Ritchey & Paine, 2018*). As a result, only 11% of the top-publishing authors in this field are females (*Maas et al., 2021*).

Publishing articles is the primary way researchers can receive recognition for their contributions and advance their careers through grants and academic positions (*Fox, Ritchey & Paine, 2018*). The underrepresentation of females in authorship patterns may therefore reflect early withdrawals of females from the academic community and also highlight the potential for implicit biases that favor males (*Sidhu et al., 2009*; *Martin, 2012*; *Salerno et al., 2019*). Indeed, female scientists have a 19.5% higher risk per year of leaving academia, and their academic careers tend to be shorter than those of their male counterparts (*Huang et al., 2020*). This results in a significant cumulative advantage for males in publishing over time (*Huang et al., 2020*). To address this issue, strategies to reduce the early dropout of female scientists from their academic careers and to decrease gender biases in the peer review process are crucial for increasing diversity, inclusion, and representativeness in scientific publications (*Huang et al., 2020*; *Diele-Viegas et al., 2021*). Thus, those strategies will ultimately prevent the underuse of valuable skills in the pool of females and other minority scientists, such as Latin Americans (*Sheltzer & Smith, 2014*; *Valenzuela-Toro & Viglino, 2021*). One possible solution to avoid explicit bias in article publication is implementing a double-anonymized peer review process.

In the double-anonymized peer review process (hereafter: DA peer review), author and reviewer identities are concealed. This peer review has been proposed as an efficient strategy to reduce biases against gender, institutions, country of origin, new ideas, authors' prestige, and young scientists (*Stensrud & Brooks, 2005*; *Mainguy, Motamedi & Mietchen, 2005*; *Smit, 2006*). Furthermore, DA peer review is seen as a more equitable peer review system (*Smit, 2006*) that promotes fairness by creating a system concealing identities and forces reviewing focused on content (*Wenneras & Wold, 1997*; *Stensrud & Brooks, 2005*; *Darling, 2015*). However, the implementation of DA is still not widespread among scientific journals (*Haffar, Bazerbachi & Murad, 2019*). DA peer review has concerns related to the additional journal workload and the possibility of reviewers identifying authors or institutions. In addition, the effectiveness of DA in reducing gender bias in publishing remains controversial. While in the fields of medicine and economics, no significant differences have been found between DA and SA peer review systems, with female authors achieving similar publication rates under both scenarios (*Blank, 1991*; *Cho et al., 1998*; *Justice et al., 1998*; *Mahajan et al., 2021*), DA has been found to effectively reduce gender bias in computer science (*Tomkins, Zhang & Heavlin, 2017*). In the EcoEvo field, DA has positively impacted the number of female authors in journals such as behavioral ecology (*Budden et al., 2008*), but this increase in female authorship may be attributed to the overall female representation (*i.e.*, relative number and proportion of publications) in the field over time (*Holman, Stuart-Fox & Hauser, 2018*; *Salerno et al., 2019*), rather than the peer-review process *per se* (*Webb, O'Hara & Freckleton, 2008*). For example, the biological conservation journal utilized a SA peer review and did not find evidence of gender bias in publication rates (*Primack et al., 2009*). Additionally, the proportion of female authors did not differ between the DA and SA journals of ecology and ornithology (*Cox & Montgomerie, 2019*). While these studies suggest comparable levels of female authorship in both DA and SA journals, the analyses were based on a few journals, which may not accurately reflect the overall publishing landscape in the field of EcoEvo.

Here, using a comprehensive dataset from 18 journals encompassing different thematic areas within ecology and evolution (*e.g.*, animal behavior, applied ecology, behavioral ecology, botany, conservation, ecosystem sustainability, evo-devo, freshwater ecology, and zoology), we investigate the impact of the DA peer-reviewing process on the gender authorship patterns of EcoEvo scientific journals. Specifically, we hypothesize that the frequency of publications with female-leading authorships is higher under a DA peer-reviewing policy when compared to the SA policy (H1). Moreover, we hypothesize that adopting the DA peer-review policy has increased the frequency of female-leading articles on time in journals that changed their peer-review process (H2).

## METHODS

### Journal selection and peer-review information

We selected scientific journals listed in the journal citation reports 2020 of the Web of Science database (WoS) that were classified within the wide ecology and evolution field. We restricted our search for journals with the 2020 impact factor (IF) equal to or higher than one, according to the Clarivate WOS database (https://jcr.help.clarivate.com/). From the 142 journals recovered by our search, we then selected journals that strictly included the following words in their scope: "Ecology and Evolution", "Ecology", "Evolution", and "Evolutionary Biology". This preliminary search yielded 135 selected journals (Appendix S1). However, as we obtained nine DA journals, we then randomly selected nine SA journals using the function "sample" implemented in the R environment, resulting in 18 analyzed journals (Table 1). Among journals presenting the double-anonymized peer-review policy, only two always adopted the DA peer-review, and seven had changed their policy from SA to DA in a specific year (see Table 1).

The definition of the peer-review policies adopted by journals as double or single-anonymized was made by checking their author information/guidelines. When this information was unavailable on the journals' website, we emailed the editorial office or editor-in-chief. For DA journals, we also gathered information on when this peer-review system was implemented. We kept only journals with at least 5 years of DA system so that the possible effects of this change could be discernible.

Owing to the huge volume of articles published annually, we limited our data collection to the first journal issue of each evaluated year. We considered the period between 2016–2020 for SA (*i.e.*, journals that always have been single-anonymized) and DA (*i.e.*, those journals that always have been double-anonymized) peer-reviewed EcoEvo journals (Table 1). For those journals that changed their peer-reviewing from SA to DA over time (hereafter: switched-review journals), we selected articles from 5 years before the switch to DA peer-review (*i.e.*, "pre" in Table 1) and from the subsequent 5 years, 2 years after (time lag) the switch (*i.e.*, "post" in Table 1). The 2-year time lag after the implementing the new peer-review policy was considered to account for any potential delays in the reviewing process (*Nguyen et al., 2015*; *Forti, Solino & Szabo, 2021*). In general, the process from initial article submission to final publication in a journal typically takes 1 or 2 years (*Fox, Ritchey & Paine, 2018*). By extracting "pre" and "post" DA adoption data, we were able to

**Table 1 The 18 randomly selected EcoEvo journals for the present study.**

| Journal (acronym) | Impact factor (2020) | Peer-review policy | Year of double-anonymized review adoption | First-year issues period | N |
|---|---|---|---|---|---|
| Conservation letters (CL) | 6.766 | Single-anonymized (SA) | – | 2016–2020 | 85 |
| Journal of applied ecology (JAnpE) | 5.840 | Single-anonymized | – | 2016–2020 | 147 |
| Journal of animal ecology (JAnE) | 4.554 | Single-anonymized | – | 2016–2020 | 104 |
| EvoDevo | 2.146 | Single-anonymized | – | 2016–2020 | 131 |
| Biological journal of the linnean society (BJLS) | 1.961 | Single-anonymized | – | 2016–2020 | 84 |
| Journal of plant ecology (JPE) | 1.833 | Single-anonymized | – | 2016–2020 | 86 |
| Australian journal of botany (AJB) | 1.386 | Single-anonymized | – | 2016–2020 | 38 |
| Journal of freshwater ecology (JFE) | 1.239 | Single-anonymized | – | 2016–2020 | 59 |
| Entomological science (ES) | 1.074 | Single-anonymized | – | 2016–2020 | 100 |
| Avian conservation and ecology (ACE) | 2.541 | Double-anonymized (DA) | – | 2016–2020 | 92 |
| Ecosystem health and sustainability (EHS) | 2.315 | Double-anonymized | – | 2016–2020 | 23 |
| Conservation biology (CB) | 5.405 | Switched-review* | 2014 | Pre**: 2009–2013/Post: 2016–2020 | 283 |
| The American naturalist (AN) | 3.744 | Switched-review | 2015 | 2010–2014/2017–2021 | 135 |
| Mammal review (MR) | 2.804 | Switched-review | 2009 | 2004–2008/2011–2015 | 60 |
| Behavioral ecology (BE) | 2.761 | Switched-review | 2001 | 1996–2000/2003–2007 | 226 |
| Journal of evolutionary biology (JEB) | 2.72 | Switched-review | 2016 | 2011–2015/2018–2022 | 173 |
| Animal behaviour (AB) | 2.689 | Switched-review | 2009 | 2004–2008/2011–2015 | 281 |
| Plant ecology & diversity (PED) | 1.196 | Switched-review | 2008 | 2003–2006/2010–2014 | 124 |

**Notes:**
The dashes refer to non-collectible data, such as DA and SA journals that follow this peer-review system from the beginning. N: the total number of articles analyzed from each journal.
*Switched-review: journals that changed their review model from single-anonymized to double-anonymized through time.
**Pre and Post: 5 years before (pre) and 2 years (time-lag change) after (post) journal adopting the DA peer-review policy.

evaluate if EcoEvo journals' adoption of a DA peer-review policy has led to an increase in the representation of female researchers as leading or senior authors. The first-year issues of the switched-review journals covered the period from 1996 to 2022 (Table 1).

We excluded editorial articles to avoid potential biases since most authors in this category are the journals' editors or previously invited authors. All other article categories submitted to the peer-review process were included in the analyses, such as commentaries, data articles, forums, and reviews (Appendix S1).

## Data

We obtained information about the first and last (senior) author's names (*i.e.*, leading authors), the number of authors, and the author's country affiliation to each article. The data was scraped from the websites of the EcoEvo journals using the *rvest* package (*Wickham & Wickham, 2016*). We manually classified the leading authors according to their gender (male/female) by performing exhaustive searches of the author names on

publicly available individual web pages and social media (*i.e.*, *LinkedIn, ResearchGate, and Twitter*), scientific platforms such as *Scopus* and *Google Scholar*, and institutional databases that include gender pronouns. When the author's pronouns were not available, we predicted their binary gender based on their given names' conventional gender. Although we acknowledge that a binary gender classification excludes several gender identities, the lack of information concerning non-binary gender diversity in the academic environment precluded the inclusion of those identities in our analyses.

## Statistical analysis

To assess the impact of the DA peer-reviewing process on the number of female-leading articles, we ran four generalized linear mixed-effect models (GLMMs) using the binomial family and logit link function (*Bolker et al., 2009*). Each article was treated as a single data point in these analyses (*Fox, Ritchey & Paine, 2018*). To test whether the frequency of publications with female-leading authorships is higher under a DA peer-reviewing policy when compared to the SA policy (H1), we fitted two GLMMs in which we considered the gender (0/1; male and female, respectively) of the first author (first model) or the senior author (second model) of each article as the response variable and peer-review policy (DA or SA) as the predictor variable. To test if adopting the DA peer-review policy has increased the frequency of female-leading articles over time in journals that changed their peer-review process (H2), we then fitted other two GLMM models. For these models, we only considered the data for the seven selected journals that have changed the peer-review policy over time (*i.e.*, switched-review journals, see Table 1). The response variable was the gender of the first (third model) or senior author (fourth model), and the predictor variable was the publication period. The publication period was classified as "pre" or "post" about a peer-review policy switch event.

As the variables time (year of publication) and the number of authors may affect the female authorship patterns (*Webb, O'Hara & Freckleton, 2008*; *West et al., 2013*) with the proportion of women in all authorship roles (except sole authorship) increasing year-on-year in EcoEvo field (*Fox, Ritchey & Paine, 2018*), we used them as fixed covariables in the models. We treated the year as a progression over time and considered it as a continuous variable (*Fox, Ritchey & Paine, 2018*). Geographic and journal (*e.g.*, article sample size and impact factor, Table 1) idiosyncrasies also directly impact gender bias in science (*Holman, Stuart-Fox & Hauser, 2018*) a pattern also observed in some EcoEvo journals (*Fox, Ritchey & Paine, 2018*; *Forti, Solino & Szabo, 2021*; *Maas et al., 2021*). Therefore, we included the country of affiliation of authors and journals as random terms (varying intercepts) in all models (*Webb, O'Hara & Freckleton, 2008*; *Fox, Ritchey & Paine, 2018*).

We performed the GLMMs using the "glmer" function from the 'lme4' package (*Bates, Maechler & Bolker, 2012*). To assess the assumptions for binomial GLMMs, we utilized the "testResiduals" function from the 'DHARMa' package (*Harting, 2020*). We quantified the goodness-of-fit of the models calculating the marginal ($R2m$) and conditional ($R2c$) coefficients of determination (*Nakagawa & Schielzeth, 2013*) using the function "r. squaredGLMM" from the package MuMIn (*Barton, 2013*). All analyses were carried out in R (*R Core Team, 2019*).
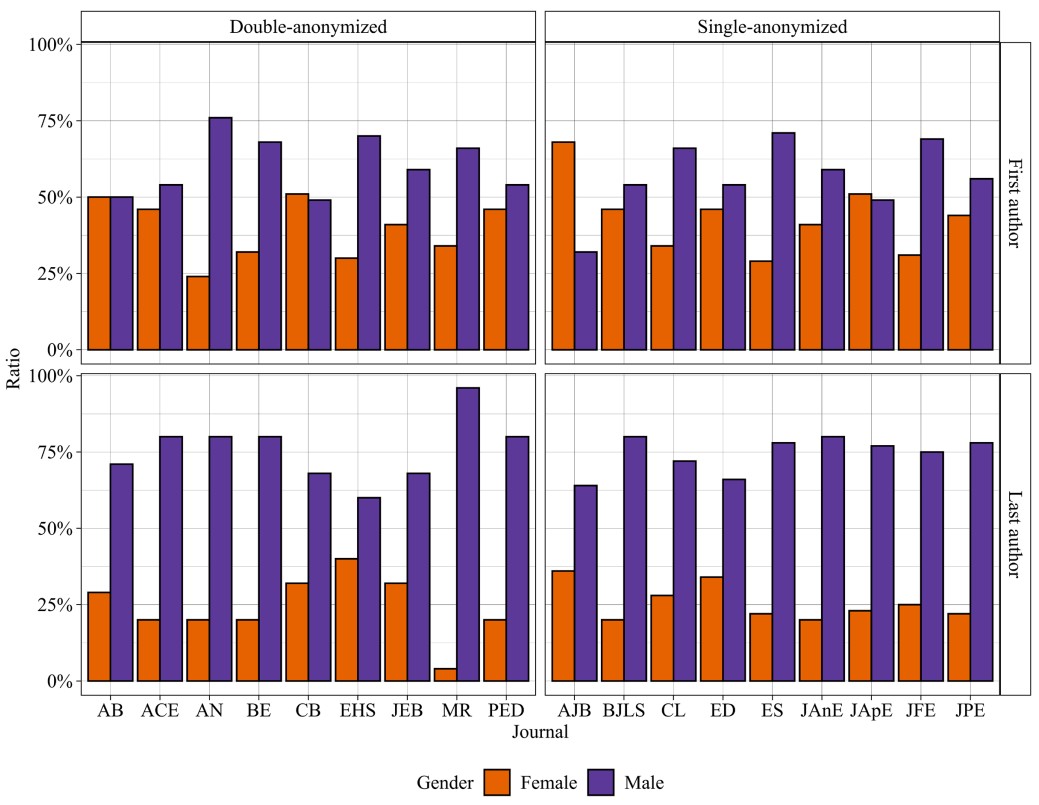

**Figure 1  Proportion of females and males as first and last (senior) authors in the double-anonymized (DA) and single-anonymized (SA) journals.**

## RESULTS

We obtained articles from the 18 selected EcoEvo journals (Table 1). After excluding the editorials ($N = 41$), our dataset resulted in 2,231 analyzed articles (Appendix S1). Overall, males led the majority of articles in EcoEvo journals, regardless of the peer-review policy and especially in the senior author position (Fig. 1). Only 24% of senior and 40% of the first authors were women in all analyzed articles.

The gender ratio difference cannot be attributed to the peer-review policy of EcoEvo journals. We did not find a significant difference between DA and SA peer-review policies concerning women publications as first or senior authors (Tables 2 and 3, respectively). Also, we did not find a significant difference between the "pre" and "post" periods (Fig. 2) in journals that switched their peer-review policy from SA to DA regarding both first and senior gender authorship (Tables 4 and 5, respectively).

## DISCUSSION

Using a comprehensive dataset of 2,231 articles from 18 journals encompassing different thematic areas of ecology and evolution, we showed that despite the increasing efforts to avoid gender imbalance in science, it is still pervasive in the EcoEvo field. We found that females were not likely to publish more in DA peer-reviewed journals (H1) and adopting

**Table 2** Generalized linear mixed model (GLMM) results for 1,593 articles from 18 ecology and evolution scientific journals.

| Fixed effects | Estimate | SE | Z | *P*-value |
|---|---|---|---|---|
| (Intercept) | −0.448 | 0.152 | −2.937 | **0.003** |
| Single-anonymized peer-review | 0.038 | 0.185 | 0.210 | 0.833 |
| Year of publication | 0.086 | 0.091 | 0.950 | 0.342 |
| Number of authors | 0.007 | 0.054 | 0.136 | 0.892 |
| **Random effects** | **Variance** | **Std. dev.** | | |
| Journal | 0.076 | 0.276 | | |
| Country | 0.123 | 0.351 | | |

Note:
Gender (male and female; *i.e.*, 0 and 1, respectively) of the first author as the dependent variable and peer-review policy, *i.e.*, double-anonymized (DA) or single-anonymized (SA), as the independent variable. We also included the year of publication and the number of authors as fixed covariables, whereas authors' country affiliation and journal as random factors in the model. Significant *P* values are in bold. The model's $R^2$ values were: marginal $R^2$ ($R^2m$) = 0.02 and conditional $R^2$ ($R^2c$) = 0.06.

**Table 3** Generalized linear mixed model (GLMM) result for 1,487 articles from 18 ecology and evolution scientific journals.

| Fixed effects | Estimate | SE | Z | *P*-value |
|---|---|---|---|---|
| (Intercept) | −1.167 | 0.135 | −8.641 | **0.000** |
| Single-anonymized peer-review | −0.061 | 0.155 | −0.394 | 0.693 |
| Year of publication | 0.153 | 0.081 | 1.887 | 0.059 |
| Number of authors | −0.228 | 0.082 | −2.787 | **0.005** |
| **Random effects** | **Variance** | **Std. dev.** | | |
| Journal | 0.020 | 0.142 | | |
| Country | 0.098 | 0.313 | | |

Note:
Gender (male and female; *i.e.*, 0 and 1, respectively) of the senior author as the dependent variable and peer-review policy, *i.e.*, double-anonymized (DA) or single-anonymized (SA), as the independent variable. We also included the year of publication and the number of authors as fixed covariables, whereas authors' country affiliation and journal as random factors in the model. Significant *P* values are in bold. The model's $R^2$ values were: marginal $R^2$ ($R^2m$) = 0.01 and conditional $R^2$ ($R^2c$) = 0.05.

the DA peer-review policy did not increase the representativeness of female researchers as first or senior authors over time in EcoEvo journals (H2).

Adopting a DA peer-review process is usually considered a possible solution to reduce the gender gap in academic publications (*Budden et al., 2008*). However, our results showed that this policy alone is not reducing the gender gap in EcoEvo journals. In fact, our analyses showed that a SA policy produced a slight effect on reducing gender imbalance.

Although the nine journals included here as DA or switched-review claim to follow a DA peer-review process, their actual implementation is not guaranteed. For example, the DA is optional in *The American Naturalist* and *Animal Behaviour*, meaning that the authors can opt between hiding or showing their identities to the reviewers. In the *Journal of Evolutionary Biology*, the reviewers' names can be shared with authors if they sign their reviews. *Mammal Review* indicates in their authors' guideline section that papers with conflicts of interest may not be suitable for DA, while *Behavioral Ecology* informs that the

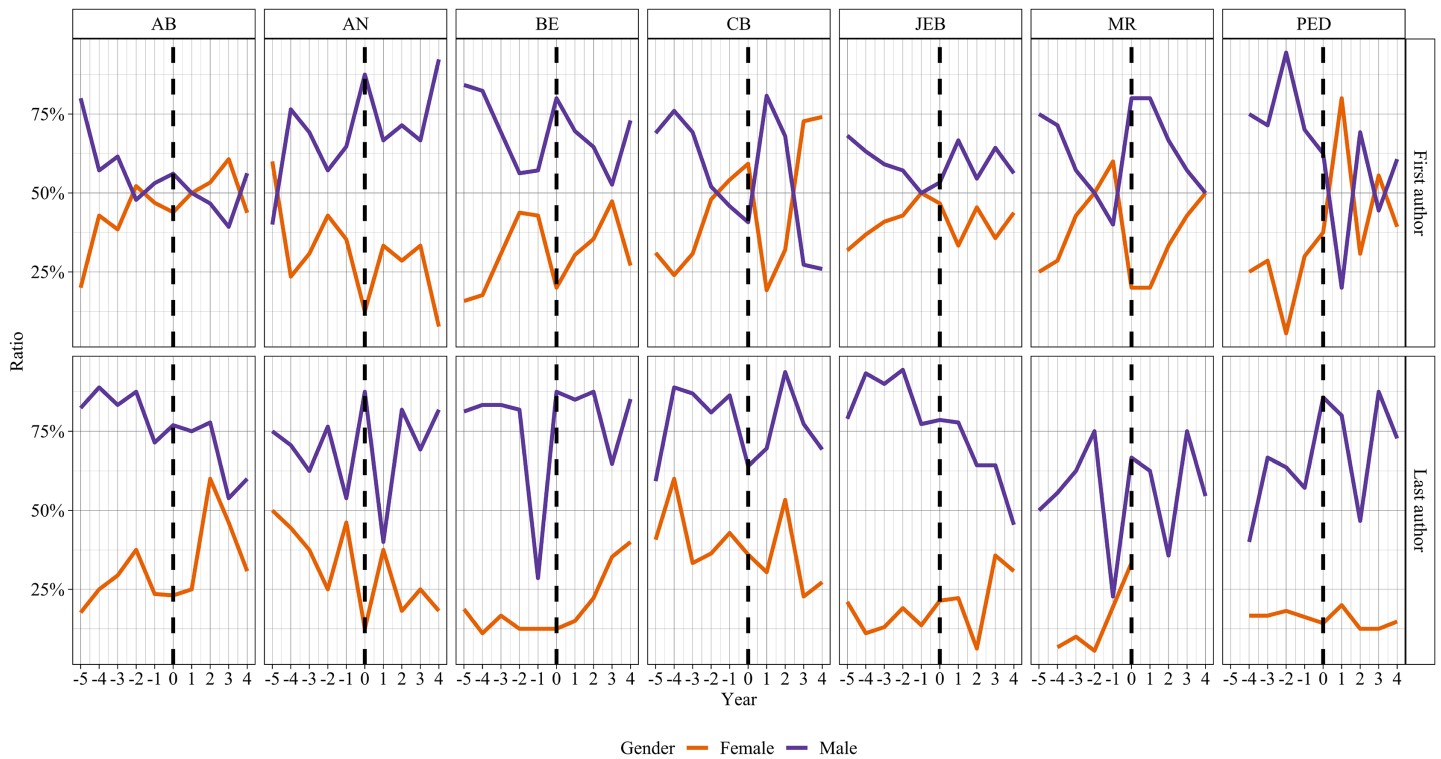

**Figure 2 Proportion of females as first and last authors in journals that switched their peer review process from single-anonymized (SA) to double-anonymized (DA) over time.** The dotted lines indicate the year of switching. Negative numbers on X-axis indicate the years when the journals were SA, and positive numbers indicate the years of the DA peer review process.

author's identity is "as much as possible" kept from reviewers. These uncertainties highlight the difficulties in actually implementing the DA peer review, which might be affecting our results.

Preparing and reviewing a DA paper demands both authors' and reviewers' specific care to ensure their anonymity (*Cox & Montgomerie, 2019*). Reviewers were able to identify the authors in 40% of the performed DA reviews in medicine journals (*Cho et al., 1998*; *Justice et al., 1998*), and 10–26% in computer science journals (*Le Goues et al., 2018*).While adopting a DA policy could reduce nepotism, and geographic, institutional, and gender biases (*Budden et al., 2008*; *Cox & Montgomerie, 2019*; *Fox & Paine, 2019*), such improvements are probably linked to a well-implemented DA that guarantees a masked review (*Cho et al., 1998*; *Darling, 2015*).

The lack of gender equality in the authorship of papers published in DA journals could also indicate that conscious and unconscious gender biases are still pervasive in the peer-review process, which may be related to the gender bias in editorial boards of EcoEvo journals (*Liévano-Latorre et al., 2020*). Female editors represent less than 30% of editorial boards in academic journals (*Liu et al., 2023*), and this trend is similar in ecology and conservation journals (*Cho et al., 1998*; *Liévano-Latorre et al., 2020*; *Sperotto et al., 2021*). As male scientists support other male scientists (*Moss-Racusin et al., 2012*), editors could bias the peer review process by rejecting submitted articles with female authorships (*Brodie et al., 2021*). On the other hand, female editors tend to accept more

**Table 4 Generalized mixed model (GLMM) results for 1,279 papers from seven ecology and evolution scientific journals have changed the peer-review policy from single-anonymized to double-anonymized.**

| Fixed effects | Estimate | SE | Z | *P*-value |
|---|---|---|---|---|
| (Intercept) | −0.521 | 0.139 | −3.741 | **0.000** |
| Period_Pre | 0.033 | 0.183 | 0.184 | 0.854 |
| Year of publication | 0.252 | 0.124 | 2.038 | **0.041** |
| Number of authors | 0.113 | 0.059 | 1.911 | 0.055 |
| **Random effects** | **Variance** | **Std. dev.** | | |
| Journal | 0.038 | 0.195 | | |
| Country | 0 | 0 | | |

Note:
We considered the gender (male and female; *i.e.*, 0 and 1, respectively) of the first author as the dependent variable and the publication period ("Period_Pre" or "Period_Post") regarding the peer-review policy change event as the independent variable. We also included the year of publication and the number of authors as fixed covariables, whereas authors' country affiliation and journal as random factors in the model. Significant *P* values are in bold. The model's $R^2$ values were: marginal $R^2$ ($R^2m$) = 0.02 and conditional $R^2$ ($R^2c$) = 0.04.

**Table 5 Generalized Mixed Model (GLMM) results for 1,118 papers from seven ecology and evolution scientific journals have changed the peer-review policy from single-anonymized to double-anonymized.**

| Fixed effects | Estimate | SE | Z | *P*-value |
|---|---|---|---|---|
| (Intercept) | −1.204 | 0.121 | −9.876 | **0.000** |
| Period_Pre | −0.172 | 0.164 | −1.049 | 0.294 |
| Year of publication | 0.167 | 0.083 | 2.000 | **0.045** |
| Number of authors | −0.122 | 0.084 | −1.442 | 0.149 |
| **Random effects** | **Variance** | **Std. dev.** | | |
| Journal | 0.049 | 0.222 | | |
| Country | 0 | 0 | | |

Note:
We considered the gender (male and female; *i.e.*, 0 and 1, respectively) of the first author as the dependent variable and the publication period ("Period_Pre" or "Period_Post") regarding the peer-review policy change event as the independent variable. We also included the year of publication and the number of authors as fixed covariables, whereas authors' country affiliation and journal as random factors in the model. Significant *P* values are in bold. The model's $R^2$ values were: marginal $R^2$ ($R^2m$) = 0.01 and conditional $R^2$ ($R^2c$) = 0.03.

women-authored articles (*de Barros et al., 2021*). In this sense, double-anonymization could reflect the structural bias and inequalities in research institutions and universities, where tenured positions are biased towards men (*Sheltzer & Smith, 2014*; *Holman, Stuart-Fox & Hauser, 2018*; *Diele-Viegas et al., 2021*). Homophily and gender equity suggest that the inclusion of more women in leadership roles could improve the gender balance (*Amrein et al., 2011*; *Mauleón et al., 2013*; *Cho et al., 2014*; *de Barros et al., 2021*).

Journals should move towards a better implementation of the DA peer review aiming to fight intrinsic biases of the peer-review process, but the adoption of the DA review system alone could be insufficient in fostering gender equality in scientific publications. In this sense, before adopting the DA peer-review policy, the first step to promoting gender equality in EcoEvo publications is promoting gender-equitableness in these editorial boards (*Liévano-Latorre et al., 2020*). In addition, supporting affirmative actions focused

on the entry and permanence of minority groups in science, diversifying decision-makers, and dismantling mechanisms that allow sexism in academic environments are deep actions that would reduce gender bias in science (*Diele-Viegas et al., 2021*). Furthermore, more transparency in the peer-reviewing process is needed. One alternative is using an open peer-review system, which could promote reviews more accurate and courteous, reducing biases and hostile or abusive comments (*Walsh et al., 2000*; *Ford, 2013*). Another alternative could be a triple-anonymized peer-review process, where editors, reviewers, and authors are anonymized (*Brodie et al., 2021*; *Conklin & Singh, 2022*). However, both alternatives would demand adaptations and changes for journals and researchers (*Walsh et al., 2000*; *Brodie et al., 2021*).

The underrepresentation of women in leadership positions also emerges as a potential explanation for the minority of females as leading authors in scientific articles, especially as the senior author as evidenced here. Even considering that female participation in STEM and specifically in biological sciences has been increasing in the last years, females are still underrepresented in senior positions (*National Science Foundation, 2018*; *European Commission, 2019*). Female scientists receive less support in academia and harsher reviews in scientific publications, which, together with recurrent sexual and moral gender-based harassment, prevents the retention and advancement of females in STEM careers (*Leaper & Starr, 2019*; *Greider et al., 2019*; *Diele-Viegas et al., 2021*). Lack of support for female scientists is also evident in the funding and grant allocation, as women have a lower probability of obtaining a grant than men, which also contributes to lower rates of female permanence in academia (*Moss-Racusin et al., 2012*; *Wijnen, Massen & Kret, 2021*).

White male cis-gender researchers from Global North still dominate the ecology area (*Nuñez et al., 2021*) and STEM as a whole, contributing to the lack of diversity in the academic environment (*Maas et al., 2021*). Meanwhile, underrepresented researchers (*e.g.*, women, people of color, and LGBTQIA+ researchers) have different barriers to keeping and advancing in a scientific career. For instance, female researchers from Latin American countries deal with the intersection of sexism, colonialism, and racism (*Bernal et al., 2019*; *Valenzuela-Toro & Viglino, 2021*). Female Latin American scientists develop their careers in countries that invest less in STEM and present a culture that highlights male pride, besides hampering literature access and present language barriers (*Bernal et al., 2019*; *Valenzuela-Toro & Viglino, 2021*).

Besides structural challenges female researchers face in the academic environment, the COVID-19 outbreak added extra barriers to their maintenance in STEM fields. The pandemic has negatively affected the productivity, networking, community building, and well-being of women in STEM, especially mothers (*Myers et al., 2020*; *Langin, 2021*). In addition, it led to disrupted collaborations and pauses in the career progressions of female scientists, as they face challenges of remote work conflicting with caregiving responsibilities (*Hipólito et al., 2020*; *Staniscuaski et al., 2020*; *Myers et al., 2020*). Consequently, female scientists have been more isolated, losing contacts and publication chances, affecting their job stability and funding (*Gabster et al., 2020*; *Myers et al., 2020*).

All academic levels could promote a fair and equal academic environment by fostering diversity, inclusion, and affirmative measures, such as scholarships and research funding

(*Diele-Viegas et al., 2021*; *Maas et al., 2021*), especially for mother researchers (*Heidt, 2023*). Furthermore, reforms in the education system, mentoring, and academic publishing are needed to reach equality in science (*Holman, Stuart-Fox & Hauser, 2018*). For instance, creating new evaluation metrics and implementing inclusive policies, such as encouraging gender equality in the editorial boards, could reduce the gender gap in STEM fields (*Liévano-Latorre et al., 2020*; *Diele-Viegas et al., 2021*; *Sperotto et al., 2021*).

## CONCLUSIONS

Our results highlight that adopting the DA peer-review system alone could be insufficient in fostering gender equality in EcoEvo academic publications. Thus, we suggest the application of means to increase female representation on editorial boards and promotion of more transparency in the peer-review process, adopting strategies to improve the DA system, or exploring other alternatives, such as open peer-review or triple-anonymized review. Ecologists and evolutionists understand how diversity (expressed in diverse facets such as functional, genetic, phylogenetic, and taxonomic) is important to ecosystems' resilience in facing environmental changes. The question remaining is: why is it so difficult to promote and keep this "diversity" in addition to equity and inclusion in the academic environment? We thus argue that all scientists, mentors, and research centers must be engaged in promoting solutions to gender bias by fostering diversity, inclusion, and affirmative measures.

## ACKNOWLEDGEMENTS

We are very grateful to Diogo Provete, Bruno Eleres Soares, Gabriel Nakamura and one anonymous referee who substantially improved the quality of this work. We also thank Marcus Vinicius Cianciaruso to foster the discussions on gender equality in the academic environment. We also thank Kunhã Asé Network of Women in Science for promoting discussions on gender equality in the Brazilian academic environment.

### Funding

Cibele Cássia-Silva is supported by a Fundação de Amparo a Pesquisa do Estado de São Paulo (FAPESP) postdoctoral fellowship (2020/09164-0). Luisa Fernanda Liévano-Latorre is supported by the Coordenação de Aperfeiçoamento de Pessoal de Nível Superior—Brasil (CAPES)—Finance Code 001. Mariane Brom Sobreiro received a fellowship from Instituto Nacional de Ciência e Tecnologia—Ecologia, Evolução e Conservação da Biodiversidade (INCT-EECBio), supported by Fundação de Amparo à Pesquisa do Estado de Goiás (FAPEG—Chamada Pública N. 03/2021). The funders had no role in study design, data collection and analysis, decision to publish, or preparation of the manuscript.

### Grant Disclosures

The following grant information was disclosed by the authors:
Fundação de Amparo a Pesquisa do Estado de São Paulo (FAPESP) Postdoctoral

Fellowship: 2020/09164-0.
Coordenação de Aperfeiçoamento de Pessoal de Nível Superior—Brasil (CAPES): 001.
Instituto Nacional de Ciência e Tecnologia—Ecologia.
Evolução e Conservação da Biodiversidade (INCT-EECBio).
Fundação de Amparo à Pesquisa do Estado de Goiás: 03/2021.

## Competing Interests

The authors declare that they have no competing interests.

## Author Contributions

- Cibele Cássia-Silva conceived and designed the experiments, performed the experiments, analyzed the data, authored or reviewed drafts of the article, and approved the final draft.
- Barbbara Silva Rocha conceived and designed the experiments, performed the experiments, analyzed the data, prepared figures and/or tables, authored or reviewed drafts of the article, and approved the final draft.
- Luisa Fernanda Liévano-Latorre conceived and designed the experiments, performed the experiments, analyzed the data, authored or reviewed drafts of the article, and approved the final draft.
- Mariane Brom Sobreiro conceived and designed the experiments, performed the experiments, analyzed the data, authored or reviewed drafts of the article, and approved the final draft.
- Luisa Maria Diele-Viegas conceived and designed the experiments, performed the experiments, analyzed the data, prepared figures and/or tables, authored or reviewed drafts of the article, and approved the final draft.

## Data Availability

The raw data are available in the Supplemental File.

## Supplemental Information

Supplemental information for this article can be found online at http://dx.doi.org/10.7717/peerj.15186#supplemental-information.

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
