# Peer review of "Overcoming the gender bias in ecology and evolution: is the double-anonymized peer review an effective pathway over time?"

_PeerJ, doi:10.7717/peerj.15186_

## Round 0.1 · original submission · Major Revisions

I have now received three excellent reviews on your manuscript. While they all acknowledge that the work is interesting and help advance gender equality in STEM, they also pointed out some drawbacks in the sampling design (especially R3) and inference of cause from it. The timing of sampling from switched journals might also have affected your inference. R3 also pointed out that the disparity in the # of papers sampled among journals might influence results. However, this can be easily solved by adding another, crossed random effects in the GLMM.
R1, as well as I, think that your main suggestion for journals to increase the representation of female authors, of adopting triple-anonymized review due to implicit gender bias in the editorial boards, is not a direct outcome of your results.

A few more comments:

L. 103 - "computer machine science" seems weird
L. 113- botanics, not botanic
L. 115 - here *we* investigate
L. 137-8 - replace websites' information by author information
L. 217-8: delete parantesis

You don't say how you diagnosed the GLMM. This has to be done before you interpret the results. I highly recommend the R package DHARMa here

·

Basic reporting

Dear Editor and authors,

I reviewed the manuscript “Overcoming the gender bias in Ecology and Evolution: is the double-anonymized peer-review an effective pathway over time?”, which discusses the gender biases in the scientific publishing between single-anonymized and double-anonymized journals in Ecology and Evolution. The study presents important patterns that might support decision-making around peer review. The study is well-performed, using appropriate theoretical background, data sampling, and statistical analysis. I commend the effort of the authors to contribute to this important topic. Below, I provide some major and minor comments on the manuscript. I highlight comments #8, #11, and #12 as the most important ones since they address a statistical decision and one of the main suggestions on peer-review policy of the manuscript.
I hope my comments can help the Editor decide and the authors improve their manuscript. I make myself available for any further clarifications.

Best regards,
Bruno Eleres Soares, Ph.D.
University of Toronto-Scarborough

Major comments
1. The second paragraph of the Introduction (L69-L79) is vital to understand the importance of this study, and a little bit more could be done to highlight the causes and consequences of publishing. First, the linkage between the first and second paragraphs can be more explicit, highlighting that the paragraph will establish how the abovementioned causes manifested in society (L64-65) will be tied to some direct consequences (gender biases in scientific publishing). Second, it is unclear if L72-73 directly states that the gender bias observed in scientific publishing is related to homophily by male reviewers overestimating the work of male authors and underestimating the work of female authors. Therefore, I would suggest authors clarify this. Third, if the abovementioned argument is being made and taken as cause for the observed gender biases, it should be directly stated and exemplified with general trends within STEM. For example: “Such implicit biases in peer-reviewing generates an overall dominance of male-authored manuscripts… etc”.
2. L84-85: I believe the systematic pushing of female scientists outside academia is a very important point in this discussion. Huang et al. (2020; PNAS: “Historical comparison of gender inequality in scientific careers across countries and disciplines”) shows that a large portion of the publishing biases is related to an early dropout of female authors from the academic landscape. If the authors agree with this argument, maybe this importance could be highlighted here.
3. L98-99: Authors state that DA increases transparency in the peer-review process. It is unclear to me how DA promotes transparency. I suggest that the authors clarify such a statement.
4. L106-107: What do authors mean by “the increase in female authorship in Ecology journals is related to time…”? I understand it was unrelated to the type of peer review (if SA or DA), but does “related to time” means that the increase is stochastic? Is it related to other variables (like increased female representation in Ecology)?
5. L107-108: Similarly, in what sense? Do Webb et al. (2008) show no gender biases in publishing independently of the type of review process? This was not clear in L106-107.
6. L112: Authors state that “those studies” included less than five journals in their analyses. Why is this important? Does this underrepresent the publishing landscape in EcoEvo?
7. L166-167: I understand the difficulties in determining gender beyond binary classes (male and female) and when the information is not accessible/open (illustrating the necessity of increased efforts to map diversity in academia). Nevertheless, I would suggest authors disclose the associated issues of assuming one’s identity by a photograph that “suggests their gender” and how that could erase valid identities.
8. L176-177: Could authors clarify why the number of authors and year are explanatory variables? Also, do authors think that could be an interaction between Year and Peer-review type? Suppose the representativeness of female researchers in academia was low in the first years of sampling; then the gender bias would be present either in SA or DA (no effect of type). In the contrary, if we observe increased representativeness in more recent times, implicit biases could be more visible (effect of type). If there is any pattern such as this (which might be plausible), adding an interaction between variables would capture it. From what I understood, the authors used additive GLMMs and no interaction between variables.
9. L200-205: The description of the % of female-led manuscripts is related to SA and DA states of the journals but is followed by the support that these differences are not significant. I would suggest rephrasing or deleting sentences that highlight differences that are not supported statistically.
10. L233-239: Interesting point. I think it would be nice to bring again the concept of homophily (just as a more direct link with the Introduction). I would also point out that increasing female representation in editorial boards might reduce gender biases in publishing (see Barros et al. 2021; Oecologia Australis: Is Oecologia Australis promoting gender equality in its review process?), and female editors at Oecologia Australis accepted more female-led articles than man-led articles. Some questions provoke us to think about the topic: What are the authors’ thoughts on co-opting homophily biases for increasing diversity and balancing the scale? Is it fair/justifiable?
11. L243-L248: Implicit bias is a major problem in peer-reviewing (e.g., gender biases, biases favoring primary English speakers), and the triple-blind seems an adequate step to keep reducing the biases. Nonetheless, I do not think a triple-blind review is a natural conclusion from the presented results. If double-blind is not reducing the gap, the expectancy that a triple-blind would work seems to place the ongoing bias on the editors. Even if this bias is the major factor leading to this gender gap, one could say that a more transparent way of tackling such an issue would be using open reviews, which would increase the public accountability of everyone in the peer-reviewing procedure, including biased editors and reviewers providing harsh, non-constructive.
12. L249-277: The authors did a robust review of the factors subjacent to gender biases in scientific publications. I suggest authors consider including gender biases in funding allocation (Wijnen et al. 2021; Scientometrics: Gender bias in the allocation of student grants) and gender differences in having a corresponding author that is not also the lead author (Fox & Paine 2019; Ecology and Evolution: Gender differences in peer review outcomes and manuscript impact at six journals of ecology and evolution). These factors might be important ones reducing the number of papers led by female authors and should also require attention from policymakers. While I agree that we should rethink the peer-reviewing process, I believe more can be done to reduce gender biases in academia (paying liveable salaries, providing support to parents in science and caretakers) and directly by journals (offering softer deadlines and editing assistance whenever needed, changing exhaustive requirements). Therefore, the direct suggestion for a triple-anonymized review (L281-282) falls short compared to much more that should be done.

Minor comments
1. Abstract, L31-32: “Using comprehensive data on papers from 20 randomly selected EcoEvo journals…”.
2. L89-90: Authors state that most journals follow a single-anonymized review system. Could the authors provide any reference to support this statement?
3. L94-95: The point that reviewers might be able to identify authors depending on their work can be more straightforward for readers unfamiliar with the topic (e.g., since the number of biologists working on the ecology of the specific group at a specific biome/state is small, etc.).
4. L113: “Botanic” should be replaced by “Botany”.
5. L115: “…, here we investigate…”
6. L130: I did not find this information in the Supplementary Information.
7. L134: “Table 1”
8. L172: A random “t” in the middle of the sentence must be deleted.
9. L177: A random “To test whether” at the beginning of the sentence needs to be deleted.
10. L196-197: A similar sentence is already stated in the M&M (where I think it makes the most sense) but also about evaluating photographs. I would say to delete this and leave the correct information only in the M&M.
11. L217: “Botanic” should be “Botany”.
12. L218: Add the oxford comma after sustainability.
13. L231: “Behavioral Ecology scope” and the journal Behavioral Ecology? Also, there is an additional “(“ in this line.
14. L232: Not necessary to cite twice Cox and Montgomerie in this sentence.

Experimental design

No comment.

Validity of the findings

No comment.

Additional comments

No comment.

·

Basic reporting

The manuscript entitled “Overcoming the gender bias in Ecology and 1 Evolution: is the double-anonymized peer-review an effective pathway over time?” investigated a timely and important issue regarding the effectiveness of double-anonymized (DA) journal policy in the representativeness of women first and senior authors in Ecology and Evolutionary journals. I consider this study an essential contribution to evidencing that adopting DA policy is not enough to reduce the gender bias in publication representativeness in ecology and evolution journals. Based on the findings, the authors propose some guidance to increase the representativeness of women in publications. Beyond the importance of the issue approached by the authors, I consider the manuscript well written, with a clear presentation of the problem being accessed. I have some general issues and other specific points that can be found below.

Possible Confounding factors

Regarding the DA and SA journals. I was wondering if we have only DA journals in the category of DA journals. I am raising this point because even though we can find in policy journals the statement of DA, it is not guaranteed sometimes. As an example, let’s take The American Naturalist journal. They have adopted a DA reviewing process. However, it is not guaranteed, meaning the author’s identity will be guessed from some submitted documents. Not only that, but also the authors submitting to the journal might decide to opt out of DA. Therefore, even a DA journal doesn’t guarantee the DA reviewing process. So, the question is, how to ensure that DA means strictly DA? I am not sure if there are journals with strict DA policy in a way that they only accept “truly” DA submissions. If not, I am not sure if this impacts the results found in this study, especially since, as the authors mentioned in the text, the effects of DA in reducing bias are not immediate. Maybe, this is a point to be discussed briefly and points to the importance of journals adopting as soon as possible the DA policy.

Another general comment, this is more to think about but doesn’t necessarily mean a flaw or require additional analysis. It’s up to the authors, but maybe it would give more support to the findings. In the discussions, the authors mentioned the underrepresentation of women on journal editorial boards. I wondered if journals with more equitable editorial boards reflect more gender equality in publications. I am not sure if the authors' data allows for performing analysis, including the editorial board gender equitability as a factor. If so, I believe it would be interesting.

Sorry if I was unclear in my comments/suggestions or if it seems aggressive or offensive. For these reasons and to ensure a fair review, I decided to waive my anonymity.
Best,
Gabriel Nakamura


Materials and methods

Line 177: incomplete sentence

Lines 186 – 190: I am not sure why the affiliation countries were included as random effects in the models, but the number of authors and year of publication was not. Is there any evidence that the country of origin of first/senior authors is less critical in determining publication biases?

Results


Line 199: I think you meant “when” instead of “on”

Figure 1: As a matter of presentation, I would change the figure to a dispersal plot showing the values for male and female authors in each of the 20 journals evaluated. I believe that this would give a better idea about the variability among journals

Experimental design

The comments/possible improvements regarding experimental design were added to the basic report field

Validity of the findings

All reports regarding the validity of the findings were included in the general comments

Additional comments

All comments were added to the basic report section

Reviewer 3 ·

Excellent Review

This review has been rated excellent by staff (in the top 15% of reviews)
EDITOR COMMENT
The review pinpointed some important drawbacks of the manuscript that neither one of the other reviewers nor I had noticed. It helped considerably in making my decision.

Basic reporting

Thank you for the opportunity to review this manuscript. I enjoyed reading it very much, and found that there was a clear logical flow throughout. I particularly like the way hypotheses were clearly stated and tested.

Tables - I think the table with the information on how many articles came from each journal should be bumped up to the main paper, rather than the supplementary material. This information is important - you infer many of your results based on this sampling design (see more on this below).

Figures - The pie chart figure doesn't do this work justice. Two suggestions that might help illustrate your findings better: simple bar graphs/histograms for journals which are either always SA or DA, and a line graph showing change through time for journals that changed from SA to DA. I think there is a wealth of information here that is missing from the somewhat simplistic figures.

Experimental design

This is important, meaningful work. I understand the inherent difficulty of carrying out research with publications where you don't know the volume/type/authorship of papers that are submitted to journals, and only have accepted/published work to infer patterns from. However, I have a few concerns that I would like to see addressed:

1) When journal input information/statistics are not available, it becomes doubly important to ensure that the journals you pick are as similar as possible in terms of research area. First, you searched for all ecology and evolution journals, and from your list, you randomly chose journals to gather data from. However, 'The American Naturalist' and 'Ecological Economics' hardly qualify as the same type of journal. This is one example, but if you go down the list of journals you selected in Table 1, I think you will find that there are several journals that look like they don't belong in this list. This is important because these journals may receive papers that are very different, and these fields may have existing patterns where male scientists submit relatively more/on par with female scientists. With your current list of journals, we cannot rule out the fact that inherent differences in what the journals received is leading to the lack of a significant pattern between single and double anonymous journals.

I would suggest non-randomly selecting your journals instead. Use impact factor and journal scope to find similar journals (a paired design may work well too), to ensure that journals are as similar at the baseline and understand whether DA truly has had no effect on female authorship.

2) For switched-review journals, you collected data right after the switch to DA from SA. Aren't there likely to be many articles in the pipeline when the switch took place? Several articles must have been submitted during the SA phase, and published during the DA phase. You might want to account for a lag of a year or two before you assess the effect of DA on female authorship to ensure that your dataset is clean.

3) There is great disparity in your sampling design. I understand that you sampled from the first issue of the year, but Animal Behavior has 601 articles sampled, while Ecosystem Health and Sustainability has a mere 43 articles. Aren't any results you see likely going to be heavily influenced by the journals from which the majority of the samples came from? This ties in to my other point about how different these journals are - if journals that are very different from others in your list are weighted more heavily, then the inferences you draw from these results are not likely to be correct.

Validity of the findings

I would suggest placing more emphasis on the difference between what your results mean for first and last female authors. You do this to some degree in the discussion, but providing more context for it from existing patterns and trends may be helpful, especially when you've changed the way you sample.

Additional comments

Line 130 - why only 10? If you mean that only 10 were always DA, then please make that clearer here.
In your sampling design, because you chose randomly, this group is represented only by 4 journals, while there are 10 SA journals. This is a large variation in sampling effort, and needs to be addressed.

Line 177 - there seems to be a typo here

---

## Round 0.2 · Minor Revisions

I have now received back the comments by the same reviewers from the previous round. Like them, I'm very satisfied with the way authors dealt with comments and critiques. Most of the issues with the interpretation of results and extrapolations were addressed. However, a few minor issues remain, as R1 and R2 pointed out. I invite authors to address those in a final revision and then resubmit.

·

Basic reporting

Dear Editor and authors,

I reviewed the updated version of this manuscript and the point-by-point response letter. The authors made all the suggested modifications or appropriately justified unperformed suggestions whenever necessary. I believe this version is appropriate for publication in PeerJ and advances the discussion on publishing and equity in academia.
I still have a single minor suggestion, though. Authors state (L115-116) that “DA peer review (…) promotes transparency in the peer review process, where articles are evaluated solely on their content.” I might be completely wrong, but to my understanding, DA peer review tries to promote fairness by anonymity, not transparency. Transparency would be being open about the full reviewing process to authors/readers, which could even be tied to SA/DA peer reviewing (publishing reviews and identities alongside the published materials) or even during the peer review (open reviews). Transparency is more about accountability than fairness (see Bianchi and Squazzoni 2022; Science and Public Policy). My direct suggestion is just for authors to cut out the “DA promotes transparency” bit, leaving the “DA promotes fairness by creating a peer review system that conceals identities and forces reviewing focused on content” part.

I hope my comments can help the Editor decide and the authors improve their manuscript. I make myself available for any further clarifications.

Best regards,
Bruno Eleres Soares, Ph.D.
University of Toronto-Scarborough

Experimental design

NA

Validity of the findings

NA

Additional comments

NA

·

Basic reporting

General comment

The reviewed version of the manuscript entitled “Overcoming the gender bias in Ecology and Evolution: is the double-anonymized peer review an effective pathway over time?” is, in my opinion, a clearer and improved version of this important and timely study. I want to thank the authors for their time in putting together this data and results that help set the ground for a necessary discussion about gender bias in Ecology and Evolution.

I have only minor suggestions that don’t compromise any findings. There are more ideas to discuss in this study or future opportunities if the authors find them relevant. The first commentary is more technical, but I believe it could improve the presentation of the results and allow a deepening of the discussion.
The second is more of a ‘thinking out loud’ kind of commentary, so feel free to disregard it if it doesn’t make sense or add anything to the discussion.

I hope my commentaries don’t seem offensive/aggressive. I am open to further discussions if needed.

Gabriel Nakamura

Regarding response #13

Thank you for your response. I think the confusion was because of the use of the word ‘covariates’ to indicate factors included in the random component of the model. As you stated in “… we use them as covariables…”. The word ‘covariables’, if not explicitly stated if fixed or random, is frequently used to refer to fixed factors included in the analysis. That’s the reason for my commentary. I know it’s a bit of a grammar-wise type of issue, but it may cause some misunderstanding. Along those lines, another thing is related to results reporting. In the tables, the authors didn’t separate what is fixed and what is the random part of the model. Maybe it should be great to make this differentiation and report a pseudo-R2 for the whole model and both parts (fixed and random). Again, I’m lining towards the statistical issue, but I believe that reporting those components will help avoid misunderstandings and allow a complete evaluation of the results.



Results

This is just a suggestion/thought.
Even after not finding a significant relationship between journal policy and gender ratio, it would be nice to emphasize the parameter values obtained from the models in the discussion. I got curious to know more about the effects obtained in the models. For example, I guess that the authors used an effects parametrization in their models (instead of a mean parametrization strategy). In this case, the intercept in table two represents the mean estimate for the Double-Anonymized policy. If that is right, a single anonymized policy produced a slight effect on reducing gender imbalance. I understand that, statistically, the differences are not enough. Still, it seems, curiously, single anonymized presents lower differences in gender participation than double anonymized (looking at the bar charts and interpreting the coefficient reported in table 3).
A possible explanation is that double-anonymization reflects the structural bias and inequalities in research institutions and universities, where professor positions are biased towards men. Given that, I was wondering if an active role in the journals would be important to reduce gender bias. I would say yes, and this active position can not be achieved by adopting DA since it will reflect the already biased academic environment. Anyway, I’m just letting some ideas fly out. This can be used as some additional discussion, not necessarily in this study.

I really like the conclusion. Specifically, I like the proposal of implementing open peer-review processes more often. I believe that this goes more toward the idea of actively trying to reduce gender bias in science. I like the emphasis on the argument that DA is far from being the solution. The bottom line is: To change the status quo and structural bias, we need to go beyond the ordinary (in this case DA). This point is well presented in the text. Thanks for this important study and all the work the Kuña Asé has been doing in bringing to light the flaws that affect women in academia. Science is more robust when we uncover its problems.

Experimental design

All my comments were reported as basic reports

Validity of the findings

all my comments were reported as basic reports

Additional comments

All my comments were reported as basic reports

Reviewer 3 ·

Basic reporting

No comment

Experimental design

I found that my concerns with regard to experimental design had been adequately addressed.

Validity of the findings

The interpretation of the results based on changes to the design was sound.

---

## Round 0.3 · accepted · Accept

Thank you for incorporating those final corrections to the manuscript. I believe it's now ready to be published as is. Congratulations